# Time-restricted feeding leads to sex- and organ-specific responses in the murine digestive system

**Lalita Oparija-Rogenmozere**[1,2©¤*], **Madeleine R. Di Natale**[1,2©], **Billie Hunne**[1], **Ada Koo**[1], **Mitchell Ringuet**[1], **Therese E. Fazio Coles**[1], **Linda J. Fothergill**[1], **Rachel McQuade**[3,4‡], **John B. Furness**[1,2‡]

**1** Department of Anatomy and Physiology, The University of Melbourne, Parkville, Victoria, Australia,
**2** The Florey Institute of Neuroscience and Mental Health, Parkville, Victoria, Australia, **3** Gut Barrier and Disease Laboratory, Department of Anatomy & Physiology, The University of Melbourne, Parkville, Victoria, Australia, **4** Australian Institute for Musculoskeletal Science (AIMSS), Western Centre for Health Research and Education (WCHRE), Sunshine Hospital, St Albans, Victoria, Australia

© These authors contributed equally to this work.
‡ These authors also contributed equally to this work.
¤ Current Address: Department of Biomedicine, University of Basel, 4058 Basel, Switzerland
* lalita.oparija-rogenmozere@unibas.ch

## Abstract

Food intake is a key regulator of the digestive system function; however, little is known about organ- and sex-specific differences in food-driven regulation. We placed male and female C57Bl/6 mice on time-restricted feeding (TRF), limiting access to food to an 8-hour window. Food was added either at dark (ZT12) or light (ZT0) onset for 14 days. Afterwards, the feeding period was delayed by 4 hours for half the mice, and the respective TRF regime continued for another 14 days. TRF from ZT12 to ZT20 led to the highest weight gain in females and the lowest in males, while improving intestinal transepithelial resistance (TEER) in both sexes. However, it also diminished food-anticipatory gene expression of several hepatic genes, particularly in female mice. Shifting food access to ZT16 increased weight gain and reduced fasting glucose levels in male mice, while also inducing strong food-driven gene expression changes in hepatic and duodenal tissues in both sexes. Feeding during the early lights-on phase (ZT0-ZT8) caused only minor physiological changes. However, it led to an overall downregulation of hepatic and an upregulation of duodenal and gastric gene expression and blunted the food-anticipatory expression response in both sexes. Delaying feeding until ZT4 was highly detrimental, reducing TEER and further disrupting gene expression in the stomach and liver in both sexes. In contrast, at least partial restoration of food-driven gene expression was seen in the duodenum, particularly in males. These findings highlight the strong sex- and organ-specific effects of food intake time on physiological and gene expression responses. Notably, we observed a lack of alignment in gene-expression responses between the gut and liver, underscoring tissue-specific sensitivity to feeding cues.

**Data availability statement:** All relevant data are within the manuscript.

**Funding:** This study was supported by the Swiss National Science Foundation Grant No. 187739 to L.O.-R. Website: https://www.snf.ch/en Link to the specific grant: https://data.snf.ch/grants/grant/187739 The funders played no role in study design, data collection or analysis or publishing decision.

**Competing interests:** The authors have declared that no competing interests exist.

## Introduction

The ability to measure time has played such an important role in human development that clocks are among our oldest inventions. Entire scientific disciplines, such as chronometry, have been devoted to understanding timekeeping and developing new standards. And yet, our own biological clock remains incompletely understood. While progress has been made in uncovering how organisms align their internal rhythms to external cues, such as the light-dark cycle, the response to the timing of food intake is less explored. Thus, the tissue- and sex-specificity of food entrainment, as well as its molecular mechanisms, remain partially unresolved.

Self-sustaining, cell-autonomous circadian clocks are essential components of our biology [1]. Early studies revealed diurnal oscillations in body temperature [2–4], food intake [5], and organ-specific functions, such as renal excretion of water and salts [6,7]. It is now well established that the circadian clock drives rhythmic expression of approximately 43% of our protein-coding genes and over a thousand noncoding RNAs [8], regulating major physiological functions across most tissues and organs [9].

At the molecular level, the mammalian molecular circadian clock consists of a transcriptional-translational feedback loop, present in nearly every cell. The core look includes the positive regulators CLOCK and BMAL1, which heterodimerize to activate gene transcription in the negative limb, including Per1–3 and Cry1–2. These proteins form complexes and return to the nucleus to inhibit the activity of CLOCK-BMAL1 heterodimer, completing the loop [1,10–14]. Additional regulatory layers involve orphan receptors such as REV-ERBs and RORs, connecting circadian output to nutrient metabolism, especially lipid homeostasis [15,16], and post-translational modifications that fine-tune the clock output [1].

These mechanisms allow the circadian system to align with external cues or "zeitgebers" [17], such as light, food, and physical activity. The formed rhythm can also be influenced by age, race, geographical location, and sex [18–21]. The light-dark cycle is the best-researched zeitgeber, entraining the central circadian oscillator in the suprachiasmatic nucleus (SCN) of the hypothalamus. The SCN synchronizes peripheral clocks through neural and hormonal outputs, as well via regulation of body temperature. However, peripheral organs and tissues, including the gastrointestinal (further: GI) tract and liver, also possess autonomous circadian oscillators that can shift phase in response to local zeitgebers such as food intake, exercise, and drugs of abuse [22–30].

Circadian misalignment arising from conflicting zeitgebers contributes to increased risks of cardiovascular, metabolic, immune, neurological, and even psychiatric disorders [31,32]. In contrast, alignment of feeding time with the light-dark cycle improves health [33,34], and time-restricted feeding (TRF), usually confining food intake to 8–10 hours per day, can alleviate obesity and improve cardiovascular outcomes in both rodents [35–38] and humans [39–42].

While substantial progress has been made in understanding food entrainment, the field still lacks an integrative view encompassing multiple peripheral tissues, both sexes, and various time scales. Research has focused primarily on the liver, where food-driven regulation of gene expression and metabolic pathways is well

documented, even when it leads to uncoupling from SCN-driven rhythms and causes inflammatory consequences [22,43–50]. In contrast, digestive organs such as the stomach and small intestine, despite being essential for food uptake and expressing clock genes [24,51–55], remain underexplored in food-entrainment research, particularly under *in vivo* conditions.

Efforts to localize food-entrainable oscillator (FEO, drives food anticipatory activity (FAA)) in the digestive system have yet to identify specific tissue or molecular mechanism; however, it appears that FEO might not rely on canonical clock genes [56–58]. Some studies suggest involvement of GI hormones like ghrelin, whose food-driven secretion might act on central and peripheral receptors [59]. Additional food-driven responses include changes in gut microbiota, lipid profiles, inflammation, other GI hormone and short-chain fatty acid secretion, and gastric contractility [60–64], suggesting that digestive organs play a central role in food-regulated physiology.

Importantly, peripheral organs can entrain to TRF independently of SCN, with organ-specific rates of adaptation [22,51,57,65–67] with some changes persisting even after a return to *ad libitum* feeding [68]. The regulatory mechanisms are complex, involving diverse functions, gene networks and signaling pathways, that often yield contrasting results under similar conditions [68,69]. While some studies suggest that peripheral clocks play a role in coordinating food-driven responses, cooperation between multiple organs, for example, liver, muscle, and adipose tissue, also appears to be important for regulating redox, lipid, and glucose metabolism [70–73].

Another underexamined factor in food entrainment is biological sex. There is evidence that SCN morphology, neuropeptide, and clock gene expression differ between males and females [74–76], alongside with hormone secretion, circadian period and behavioural rhythms [77,78]. In the context of TRF, male mice display earlier and more pronounced FAA [79] than females and additional sex-specific differences have been reported in gene expression in kidney and heart [67]. Liver responses to feeding time appear more comparable between sexes [67], but studies focusing on hepatic post-translational regulation have only been conducted in females [80]. The exclusion of one sex from most studies continues to limit our understanding of sex-specific entrainment.

In this study, we investigated how time-restricted feeding influences food-driven, sex-specific regulation of weight gain, glucose handling, intestinal barrier function, and genes expression in murine digestive system. Male and female mice were subjected to an 8-hour TRF regime starting either at the onset of the dark (ZT12) or light (ZT0) phase, with or without a 4-hour feeding delay following 14 days of entrainment. Rather than aiming to characterize circadian rhythmicity, we focused on food-driven responses at three key physiological timepoints: food anticipation, intake, and postprandial period, and how these responses change due to the duration of entrainment. This approach enables us to detect organ- and sex-specific shifts in gene expression in relation to feeding time.

## Materials and methods

### Ethical approval

All experimental procedures and handling involving mice were approved by the University of Melbourne Animal Ethics Committee (application #1914983) and performed in compliance with the Australian code for the care and use of animals for scientific purposes.

### Experimental animal origin and housing

In total, 180 male and 180 female C57Bl/6 mice (RRID:IMSR_JAX:000664) from the Animal Resources Centre (Canning Vale, WA, Australia) were used in the study. All mice arrived at the local animal facility aged 5–7 weeks and were housed in groups of five, under a 12:12 h light-dark (further: LD) cycle, at 22 ± 1 °C, 55–60% humidity, with standard chow (Barastoc 102108, 12.8 MJ/kg, Ridley Corporation, Melbourne, Australia) and water provided *ad libitum*. After three days of post-arrival adjustment to the local facility, mice were transferred to either LEDDY cages (black cages with individual lighting systems) (Cat# GM500) or to the Aria ventilated cabinet (Cat# 9BIOC44R4Y1, both from Tecniplast, Buguggiate, Italy),

and the light onset of 12:12 h LD cycle was adjusted such that the sample collection from day-fed and night-fed mice could be done in parallel. Mice were adjusted to the new 12:12 h LD cycle and housing conditions for 1 week, based on observations that young mice re-entrain within 5–10 days, especially regarding peripheral rhythms [81] and their activity [82]. Chow and water were provided *ad libitum* during this time. After this week, mice were weighed at the end of the dark phase (Zeitgeber time (ZT) 24). Obtained values were recorded as pre-time-restricted feeding weights (further: pre-TRF) and used as a baseline for later weight gain calculations.

## Food entrainment of mice

Overall, two independent experiments were performed due to space and infrastructure limitations, with random allocation of TRF groups. No *ad libitum* fed group was used in this study. Food intake was not measured, as mice were group-housed.

After a 1-week adjustment to the new LD cycle, all mice within a cage (n = 5) were placed on the same TRF schedule for 14 days (Fig 1, Pre-shift days). In some groups, TRF was started with a 1–2 day delay to avoid overlapping sacrifice and sample collection. During TRF, mice had access to standard chow for 8 h per day, either starting at dark onset at (ZT12, "Restricted night-fed", grey bar in Fig 1) or at light onset (ZT0, "Restricted day-fed", yellow bar), followed by a 16 h fasting period. To prevent mice from consuming leftover food debris during fasting, each group rotated between two cages: a "food cage" during feeding hours and an "empty cage" with no chow during fasting period. Water was provided *ad libitum* at all times. Mice were weighed weekly at the end of their respective feeding period (Fig 1, blue arrows).

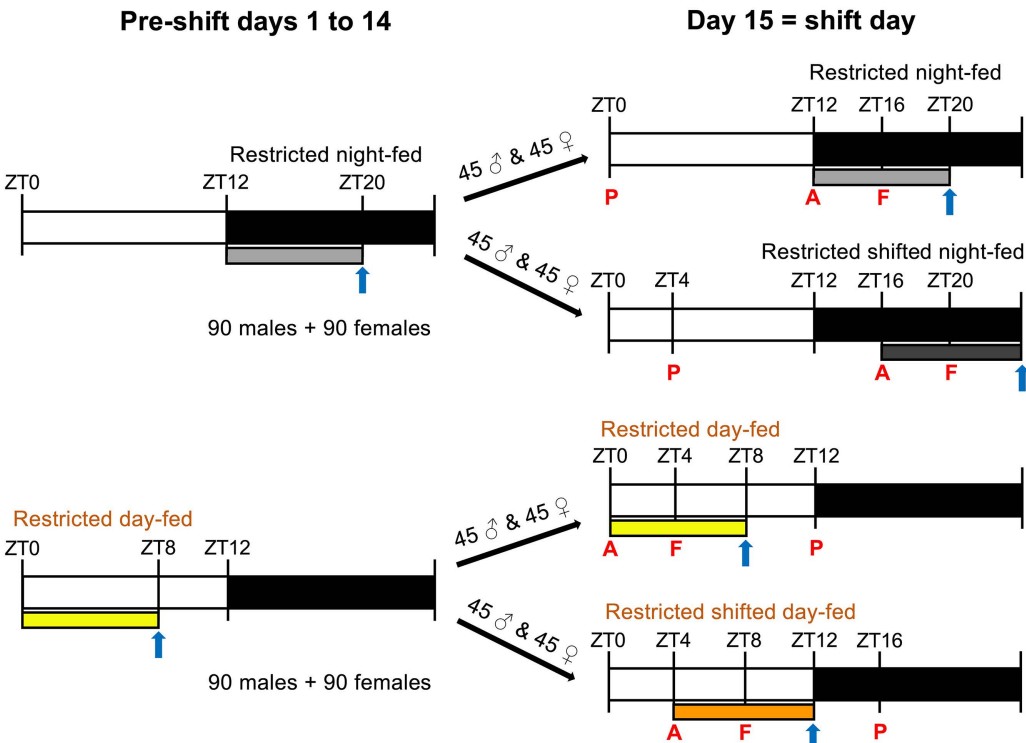

**Fig 1. Food entrainment schedule.** Bars indicate the placement and length of the feeding period. Arrows show weekly weight measurement timepoints. Mice were sacrificed on the 3rd, 7th, or 14th day after feeding shift. Letters indicate sacrifice timepoints on each day: A = food anticipation, F = food intake, P = postprandial period. n = 5 per sacrifice timepoint per sex.

After 14 days of TRF, the feeding window was delayed by 4 hours for half of the mice (Fig 1, Day 15). These new groups were designated as 'Restricted shifted night-fed' (ZT16-ZT0, dark grey bar) and 'Restricted shifted day-fed' (ZT4-ZT12, orange bar). The other half of the mice continued their original TRF. Weight gain measurements were continued as previously stated (Fig 1, blue arrows).

Mice were sacrificed on days 3, 7 and 14 after this shift, to assess food-driven responses at three biologically revelant timepoints: 1) food anticipation ("A"): after 16 h of fasting, just before the anticipated feeding; 2) food intake ("F"): 4 h into the feeding period; 3) postprandial period ("P"): 4 h after the end of the feeding window. On each sacrifice day, at each timepoint, five female and five male mice were sacrificed.

Mice were anesthetized with isoflurane (Cat# FGISO0250, Pharmachem, Eagle Farm, Australia), and euthanized by a cervical dislocation. Body weight was recorded, and the following organs and tissues were collected and snap-frozen using liquid nitrogen: the right lobe of the liver, the dorsal half of the stomach, and the duodenal mucosa. Dduodenal mucosa was collected by opening the first 2 cm of the duodenum and scraping its inner surface with a surgical scalpel at a 45-degree angle. The ileum (5 cm proximal from the cecum) was collected only during the food intake timepoint (Fig 1, "F") for measurements of intestinal permeability and TEER.

## Blood glucose measurements

Blood glucose was measured at the food anticipation timepoint (Fig 1, "A"). A drop of blood from the tail vein was applied on the Accu-Chek® Performa Test strip (Cat# 06454038020), and glucose concentration was measured with Accu-Chek® Performa (Cat# 05894964014) blood glucose meter (both from Roche Diagnostics, Manheim, Germany).

## Intestinal electrical resistance measurements

The ileum collected during the food intake timepoint (Fig 1, "F") was cut in half. Both pieces were placed in Krebs-Henseleit buffer (11.1 mM glucose, 118 mM NaCl, 4.8 mM KCl, 1.0 mM $NaH_2PO_4$, 1.2 mM $MgSO_4$, 25 mM $NaHCO_3$, 2.5 mM $CaCl_2$, pH 7.4), opened along the mesenteric border and pinned (full-thickness) onto Ussing chamber sliders (Cat# P2311, 0.3 cm² apertures, Physiological Instruments, San Diego, USA). Sliders were inserted in the middle of two-part Ussing chambers (EasyMount Diffusion Chambers, Physiologic Instruments, San Diego, USA), and 5 ml Krebs-Henseleit buffer solution was added to the serosal side of the tissue. The mucosal side of the tissue received 5 ml of modified Krebs-Henseleit buffer, where glucose was substituted with 11.1 mM mannitol. This was done to avoid apical uptake of glucose while still maintaining an osmotic balance. Buffers in both chambers were kept at 37°C and bubbled with carbogen (5% $CO_2$, 95% $O_2$) to maintain an optimal pH level. A multichannel voltage-current clamp (VCC MC6, Physiologic Instruments, San Diego, USA) was applied to each chamber through a set of four electrodes (2 voltage sensing and 2 current passing electrodes) and agar bridges (3% agarose/3 M KCl in the tip and backfilled with 3 M KCl), installed on opposite sides of the tissue. The tissue was left to equilibrate for 20 min before clamping the voltage to 0 V.

To calculate TEER ($\Omega \cdot cm^2$), 2-sec pulses of 2 mV were administered to tissue every 60 sec for 1 h, and the measured net resistance was multiplied by the surface area. Voltage and short circuit current (Isc) measurements were recorded using a PowerLab amplifier and the software LabChart® 5 (RRID:SCR_017551, both ADInstruments, Sydney, Australia).

## RNA extraction

The RNA isolation was randomized and done in small batches (24 samples) to ensure representation of various TRF groups in the same batch, as well as fast processing and good RNA quality. Total RNA from approx. 20 mg of the right lobe of the liver, dorsal half of the stomach, and scraped duodenal mucosa was extracted using ISOLATE II RNA Mini Kit (Cat# BIO-52073, Meridian Bioscience, Cincinnati, USA) with the following adjustments to the manufacturer's instructions. First, tissue pieces were immediately snap-frozen in 2 ml screw-cap tubes containing Lysing Matrix D (Cat# 6540−434, Lot# 99999, MP Biomedicals, Irvine, USA) and stored at -80°C until further processing. Second, the volume of the provided

lysis buffer was increased to 500 µl per sample, and 5 µl of β-mercaptoethanol was added. Lysis buffer was applied while the samples were still frozen and immediately followed by lysis at 6000 rpm for 2 x 30 sec in a Precellys 24 homogenizer (RRID:SCR_022979, Cat# 03119–200-RD010, Bertin Technologies SAS, Montigny-le-Bretonneux, France). The volume of RNAse-free Ethanol (Cat# EA043, Chem-supply, Gillman, Australia), used for RNA binding, was also increased to 500 µl per sample, and lysate-ethanol filtration through the provided column was done in two consecutive steps. The centrifugation time for all washing steps was increased to 1 min to ensure better removal of wash buffers. Samples were eluted in 100 µl (liver) or 60 µl (stomach, duodenal mucosa) of RNase-free water (Cat# 10977−015, Lot# 2186758, Invitrogen by Life Technologies, Grand Island, USA). The elution step was repeated using the initial eluate to increase the RNA yield. The quality and quantity of the extracted RNA were assessed using a 2200 Tape Station (RRID:SCR_014994, Agilent Technologies, Santa Clara, USA) and a Nanodrop ND-1000 UV spectrophotometer (RRID:SCR_016517, NanoDrop Technologies, Wilmington, USA), respectively.

The synthesis of cDNA was performed using 100 ng RNA in a 20 µl total reaction volume using an iScript Reverse Transcription Supermix kit (Cat# 1708841, Bio-Rad, Hercules, USA) without any changes from the manufacturer's instructions. Samples were split across 96-well plates, with 6 negative controls included on each plate, these were randomly chosen RNA samples where no cDNA synthesizing enzyme was added. Synthesis reaction was carried out using PCRExpress (Cat# 630−003, Thermo Hybaid, Franklin, USA) or Bio-Rad T100™ (RRID:SCR_021921, Bio-Rad, Hercules, USA) thermal cyclers.

## Determination of gene expression

After cDNA synthesis, 10 µl of each cDNA sample was delivered to the Translational Research Facility within Monash Health Translational Precinct to determine gene expression levels using the Fluidigm Digital Array Integrated Fluidic Circuits [83] (further: IFCs). First, all cDNA samples underwent quality control, where expression of either *Gapdh* (forward primer: TGACCTCAACTACATGGTCTACA, reverse: CTTCCCATTCTCGGCCTTG) or *β-actin* (forward primer: GGCTG-TATTCCCCTCCATCG, reverse: CCAGTTGGTAACAATGCCATGT) was tested as SYBR assays using QuantStudio 6 Flex RealTime PCR reader (RRID:SCR_020239, Thermo Fisher Scientific, Waltham, USA). Afterwards, samples underwent pre-amplification according to the manufacturer's instructions [84]. In short, all 24 TaqMan assays for genes of interest (Table 1) were first pooled and diluted in Tris-EDTA buffer (pH 8.0) to a final concentration of 180 nM per assay. Then 3.75 µl of gene assay mix was added to 1.25 µl of cDNA sample and pre-amplified for 14 cycles using Veriti™ 96-well Thermal Cycler (RRID:SCR_021097, Cat# 9902, Thermo Fisher Scientific, Waltham, USA). Pre-amplified cDNA samples were further diluted 1:5 with Tris-EDTA buffer (pH 8.0) and loaded on the 192.24 Dynamic array IFC (Cat#100–6266, multiple lots used, Fluidigm, San Francisco, USA) together with gene assays, following Fluidigm® 192.24 Real-Time PCR Workflow Quick Reference PN 100–6170 [84]. Two array plates per each organ were used, with random sample allocation. The qPCR reaction was performed using the Biomark™ HD system (RRID:SCR_022658, Cat# BMKHD, Fluidigm, San Francisco, USA).

## Data analysis and statistics

Experimental data analysis was performed using GraphPad Prism v8.2 (RRID:SCR_002798, GraphPad Software, San Diego, CA, USA). Normality testing of the data was not conducted due to the limited sensitivity of such tests with small sample sizes. However, since the group sizes and variances were comparable, we used parametric tests with additional non-parametric tests when appropriate, given their robustness under these conditions.

To demonstrate weight gain, data from each mouse within the same feeding group and sex (n = 90 for pre-TRF, week 1 and week 2; n = 30 for week 3, n = 15 for week 4) were pooled and shown as a mean group value and 95% Confidence Interval, CI. Significant differences in weekly weight gain between the night-fed group and other feeding groups were assessed by Two-way analysis of variance (ANOVA) with Šidák's multiple comparison test.

**Table 1. TaqMan Gene assays used to determine relative gene expression levels.**

| Assigned group by function | Gene | Gene accession number | Assay ID | Tested in samples from: |
|---|---|---|---|---|
| Reference gene | *18s (Rn18s)* | NR_003278.3 | Mm04277571_s1 | Liver, stomach, duodenum |
| Core clock genes | *Clock* | NM_001289826.1 | Mm00455950_m1 | |
| | *Bmal1 (Arntl)* | NM_001243048.1 | Mm00500226_m1 | |
| Glucose sensing, transport and metabolism | *Tas1r2* | NM_031873.1 | Mm00499716_m1 | |
| | *Sglt1 (Slc5a1)* | NM_019810.4 | Mm00451203_m1 | |
| | *Glut2 (Slc2a2)* | NM_031197.2 | Mm00446229_m1 | |
| | *Pepck (Pck1)* | NM_011044.2 | Mm01247058_m1 | |
| | *Gys2* | NM_145572.2 | Mm01267381_g1 | Liver |
| Amino acid and peptide transport | *Pept1 (Slc15a1)* | NM_053079.2 | Mm04209483_m1 | Liver, stomach, duodenum |
| | *B⁰at1 (Slc6a19)* | NM_028878.3 | Mm01352157_m1 | |
| Lipid metabolism | *Mttp* | NM_001163457.1 | Mm00435015_m1 | |
| | *Fasn* | NM_007988.3 | Mm00662319_m1 | |
| | *Cyp7a1* | NM_007824.2 | Mm00484152_m1 | |
| | *Pparα* | NM_001113418.1 | Mm00440939_m1 | |
| | *Srebf1* | NM_011480.3 | Mm00550338_m1 | Liver |
| Biosynthesis of NAD+ | *Nampt* | NM_021524.2 | Mm00451938_m1 | |
| Hormone signalling | *Ffg21* | NM_020013.4 | Mm00840165_g1 | |
| | *Igf-1* | NM_001111274.1 | Mm00439560_m1 | |
| | *Tph1* | NM_001136084.2 | Mm01202614_m1 | Stomach and duodenum |
| | *Sst* | NM_009215.1 | Mm00436671_m1 | |
| | *Ghrl* | NM_001286404.1 | Mm00612524_m1 | |
| | *Gast* | NM_010257.3 | Mm00439059_g1 | Stomach |
| | *Lepr* | NM_001122899.1 | Mm00440181_m1 | |
| | *Pga5* | NM_021453.4 | Mm01208256_m1 | |
| | *Gip* | NM_008119.2 | Mm00433601_m1 | Duodenum |
| | *Gcg* | AF276754.1 | Mm00801714_m1 | |
| | *Cck* | NM_031161.4 | Mm00446170_m1 | |
| Pro-/Anti- inflammatory response | *TNFα* | NM_013693.3 | Mm99999068_m1 | Liver, stomach, duodenum |
| | *Il-6* | NM_031168.1 | Mm00446190_m1 | |
| | *Il-17a* | NM_010552.3 | Mm00439619_m1 | |
| | *Crp* | NM_007768.4 | Mm00432680_g1 | Liver |
| | *Gpx1* | NM_008160.6 | Mm00656767_g1 | |
| | *Gpx2* | NM_030677.2 | Mm00850074_g1 | Stomach and duodenum |

*18s* was used as a reference gene for the calculation of relative expression ratios (R), using the following formula: $R = 2^{-(Ct(test\ RNA)-Ct(18S\ RNA))}$. All relative expression ratios were further normalized against the average value shown by night-fed males at the food anticipation timepoint (Fig 1, point "A") on the 3rd day after the feeding shift.

Measured TEER values were submitted to an outlier test, with all values outside of mean +/- 2*(standard deviation interval) excluded from further analysis. This was done to exclude tissue pieces that might have been damaged during the resistance measurements.

Weight at the time of sacrifice (Fig 2B), fasting blood glucose values, and TEER values were shown as separate data points per individual mouse, together with the mean group value and standard deviation (Further: SD). Two-way ANOVA was used together with Dunnett's multiple comparison test to determine differences between all experimental groups independently of the sacrifice day. However, only significant differences between the night-fed group and other TRF regimes

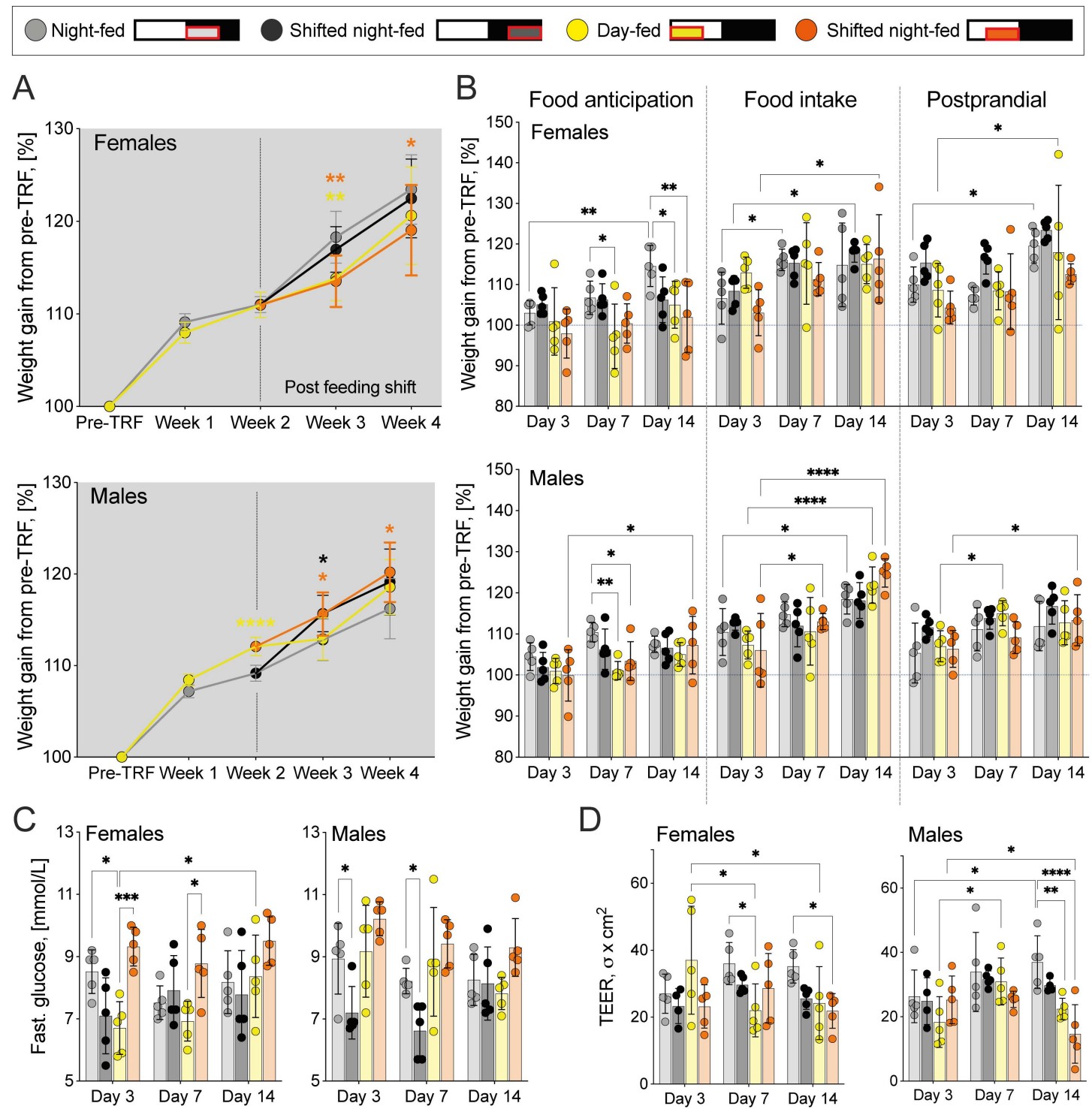

**Fig 2. Physiological effects of time-restricted feeding.** A: Post-feeding weights were normalized to individual pre-food entrainment (pre-TRF) values. Mean (95% CI) shown, n = 90 mice per sex at pre-TRF, week 1 and week 2; week 3 n = 30 mice per sex; week 4 n = 15 mice per sex. Two-way ANOVA with Šidák's multiple comparison test, showing differences between the night-fed group and other TRF regimes on each sacrifice week, *p < 0.05, **p < 0.01, ***p < 0.001, ****p < 0.0001. B: Differences in recorded sacrifice weights. Values normalized to individual pre-TRF values. C: Fasting glucose values recorded at food anticipation timepoint. D: Transepithelial electrical resistance (TEER) in the ileum, recorded at food intake timepoint. B to D:

Mean (SD) shown, n = 5 mice per sex per timepoint. Two-way ANOVA with Dunnett's multiple comparison test, showing differences between the night-fed group and other TRF regimes on each sacrifice day and differences between various sacrifice days (Day 3 as a control) within each feeding group, *p < 0.05, **p < 0.01, ***p < 0.001, ****p < 0.0001.

on each sacrifice day and the differences between values recorded on various sacrifice days within the same feeding group are shown in the figure.

To show gene expression differences between male and female mice, normalized expression ratios of mice representing the same feeding group, sacrifice timepoint, and sex were pooled together and shown as a tile within a heatmap. Fold changes in gene expression were shown as Log2 values. To further demonstrate the differences in gene expression between the food anticipation, intake, and postprandial period (=timepoint-specific differences), the nonparametric Kruskal-Wallis test was used together with Dunn's multiple comparison test (food anticipation timepoint vs food intake or vs postprandial period). The p-values were shown as: *p ≤ 0.05, **p ≤ 0.01, ***p ≤ 0.001, ***p ≤ 0.0001. Differences in gene expression introduced by the 4 h feeding shift were assessed with the same statistical test, comparing values between shifted and non-shifted groups at the same timepoint (e.g., night-fed postprandial vs shifted night-fed postprandial). The p-values were shown as: £p ≤ 0.05, ££p ≤ 0.01, £££p ≤ 0.001, ££££p ≤ 0.0001.

## Results

### Time-restricted feeding impacts weight gain, fasting glucose levels, and intestinal permeability in a sex-specific manner

To determine how feeding time affects various physiological parameters, we first recorded the baseline (pre-TRF) weights of all experimental mice (180 males and 180 females). One day later, we placed mice on TRF, starting at either ZT12 (night-fed, NF) or ZT0 (day-fed, DF), with feeding restricted to 8 h in 24 h period. After 14 days half the mice from each group underwent additional 4 h feeding delay (to ZT16 or ZT4, forming shifted night-fed, SNF and shifted day-fed, SDF groups, respectively). This approach enabled us to study time-of-feeding effects and shift-induced responses across sexes (Fig 1).

### Weight gain

The experimental weights were related to their pre-TRF values and expressed as a change in percentage (%) over time. Weight gain in the NF group was used as a control value, allowing the determination of the impact of lights-on feeding and delay shifts; moreover, this regimen is also the closest to the natural food intake pattern in mice [85].

We observed TRF- and sex-specific differences in weight gain. NF females gained the most weight, while night-fed males gained the least (Fig 2A, in grey). In contrast, feeding during the light-on phase (DF, SDF) suppressed weight gain in females, but increased it in males, especially when feeding start was shifted to ZT4 (SDF) (Fig 2A, yellow and orange). Male mice were overall more responsive to 4 h food delay shifts, with both resulting in a significant increase in weight gain. However, weight gain differences between TRF regimes in both sexes, tended to lessen over time, with only SDF mice showing significantly different weight gain by week 4.

The bodyweight was also recorded at each sacrifice timepoint: food anticipation (after 16 h fast, when mice expect to be fed, Fig 1, "A"), food intake (4 h after food was added, Fig 1, "F") and postprandial period (4 h after food was removed, Fig 1, "P") on all sacrifice days. The earliest sacrifice was performed on Day 3 after the delayed feeding shift (Day 3), followed by sacrifices on Day 7 and Day 14.

We then compared the sacrifice weights to pre-TRF values (Fig 2B, blue dashed line) and observed that DF and SDF mice, especially females, showed higher weight loss at the food anticipation timepoint, with some even reaching or

dipping below the pre-TRF weights. In male mice, this effect disappeared under longer entrainment (Fig 2B, top panel, Food anticipation). Weights at the food intake timepoint (Fig 2B, Food intake) tended to increase over time for all TRF regimes, especially in male mice.

Weights during the postprandial period (Fig 2B, Postprandial) did not differ between TRF regimes, suggesting similar rates of weight loss after 4h fasting.

### Blood glucose

Fasting blood glucose was measured at the food anticipation timepoint (Fig 1, "A"). We observed that female DF and male SDF mice showed significantly lower fasting blood glucose levels on Day 3, when compared to their NF counterparts (Fig 2C). However, these differences disappeared by Day 14 post-shift, suggesting metabolic adaptation. Interestingly, delaying the food intake until the late lights-on phase (SDF) quickly restored the blood glucose levels in female mice to NF-like values (Fig 2C, grey vs orange vs yellow).

### Intestinal barrier function (TEER)

Ileum from mice sacrificed during the food intake timepoint (Fig 1, "F") was used to determine TEER that reflects intestinal epithelial resistance. All feeding groups, independently of sex, showed similar TEER values on sacrifice Day 3 (Fig 2D, Day 3). However, TEER increased significantly in NF male mice over time, indicating improved barrier integrity under nighttime feeding. In contrast, SDF in both sexes and DF in male mice led to a significant decrease by Day 14 (Fig 2D, Day 14), implying that feeding during the lights-on phase leaves a detrimental impact on intestinal permeability.

### Feeding time alters gene expression in the digestive system in an organ- and sex-specific manner

To assess the molecular impact of TRF, we examined the expression of genes involved in the molecular clock, nutrient metabolism, and immune response in the liver, stomach, and duodenal mucosa. Tissue samples were collected from all experimental animals on sacrifice days 3, 7, and 14 (post-shift), at three physiologically meaningful timepoints: food anticipation, food intake, and postprandial period (Fig 1, "A", "F" and "P", respectively). Gene expression was first normalized against the housekeeping gene *18s* and then to the expression level in NF male mice at the food anticipation timepoint on sacrifice Day 3. Afterwards, the data from each experimental group (same sex, sacrifice day, and timepoint) was pooled and shown as a single tile within the heatmap.

Gene expression responses fell into three broad categories: (1) non-responsive, (2) timepoint-specific, and (3) shift-specific. Time-point specific changes were classified as significant changes in expression between food anticipation and one of the other timepoints within the same TRF group (shown as $^*$p). Shift-specific changes were defined as expression differences between shifted and unshifted groups at matched ZTs (shown as $^\$$p). It is important to note that our chosen sampling resolution did not allow assessment of rhythmic parameters like phase or amplitude; thus, we cannot describe circadian rhythmicity *per se*.

### Food-driven regulation of *Bmal1* and *Clock* is organ-specific

To investigate whether the timing of food intake affects the expression of core clock genes, we analysed *Bmal1* and *Clock* expression in the liver, stomach, and duodenum. We observed that only *Bmal1*, but not *Clock*, showed consistent, significant differences across food anticipation, feeding and/or postprandial timepoints in both sexes and all tested organs (Fig 3, $^*$p).

It has been shown that in *ad libitum* fed mice, *Bmal1* expression in the liver typically peaks during the late dark phase (between ZT20 and ZT0) [73]. We observed higher expression of the hepatic *Bmal1* at the food intake or postprandial

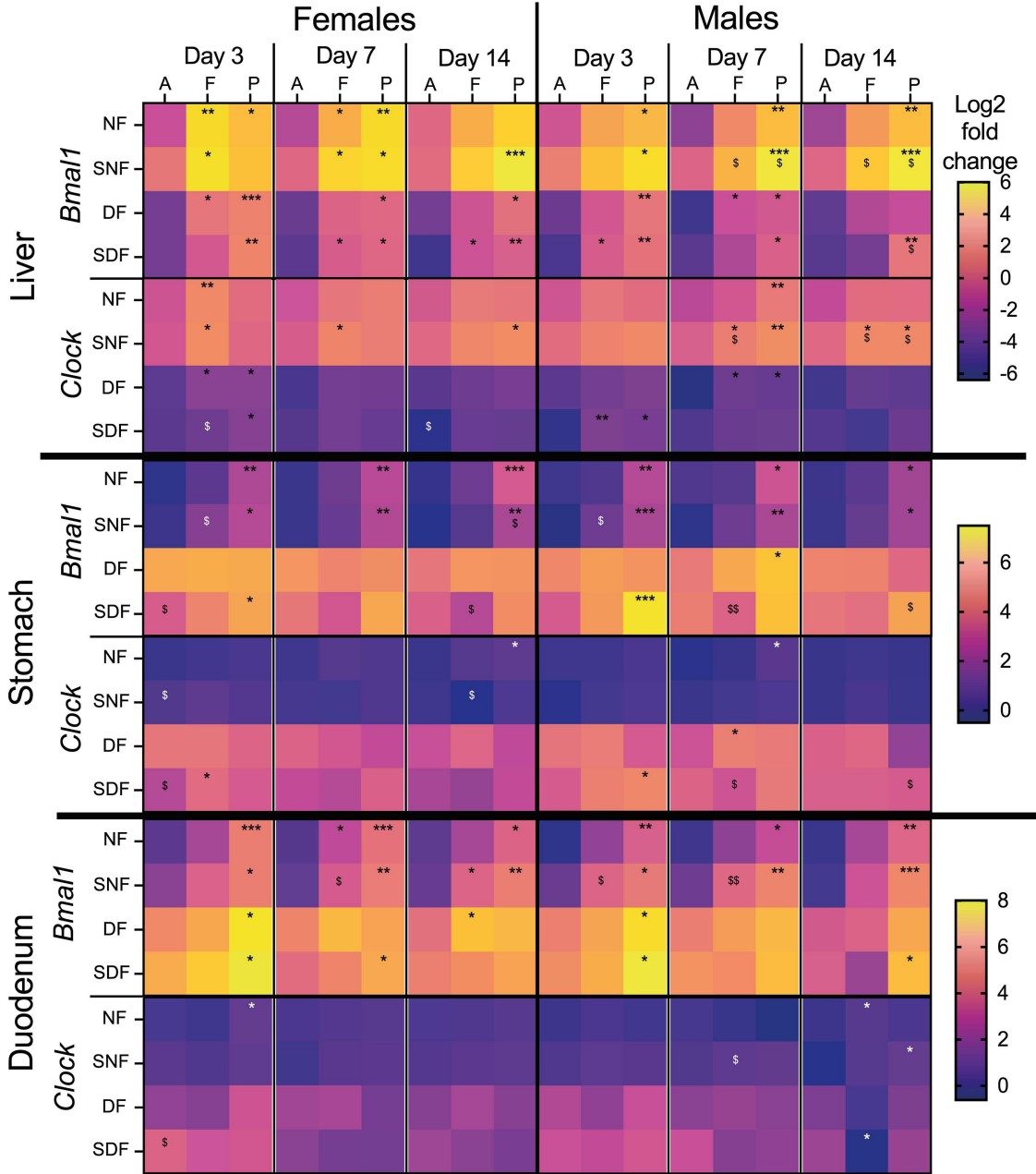

**Fig 3. Changes in *Bmal1* and *Clock* expression in male and female mice under TRF.** Sample collection timepoints shown at the top: A = food anticipation, F = food intake, P = postprandial period. Feeding conditions shown on the left: NF = night-fed (ZT12 – ZT20), SNF = shifted night-fed (ZT16 – ZT24), DF = day-fed (ZT0 – ZT8), SDF = shifted day-fed (ZT4 – ZT12). Relative gene expression values (Log2) were normalized to NF male mice on sacrifice day 3 at the food anticipation timepoint and pooled by sex, timepoint, and sacrifice day. The mean value of each experimental group is shown as an individual tile (n = 4-5 mice). Non-parametric Kruskal-Wallis test with Dunn's multiple comparison test used to show timepoint-specific (food anticipation vs intake or vs postprandial period) differences in gene expression on each sacrifice day, *p < 0.05, **p < 0.01, ***p < 0.001. The same test was used to assess shift-specific differences (non-shifted vs shifted) at each timepoint, $p < 0.05, $$p < 0.01.

timepoint under all tested TRF regimes, consistent with food-responsive regulation, as these occurred at different and sometimes opposite ZTs (Fig 1). Although some timepoint-specific responses were present for the hepatic *Clock* gene, they were less consistent.

In contrast, *Bmal1* expression in the stomach and duodenum responded poorly to daytime feeding (Fig 3). In DF and SDF groups expression levels were more uniform across timepoints or diminished over time. However, in NF and SNF mice of both sexes, *Bmal1* expression in these organs was increased at the postprandial timepoint. The *Clock* gene showed little variation across timepoints, consistent with previous findings that this gene is not strongly responsive in the stomach [53].

The 4 h feeding delay had a minimal and inconsistent effect on *Bmal1* or *Clock* expression, with only occasional differences appearing between shifted and non-shifted groups (Fig 3, differences shown as $^\$$p). Notably, DF and SDF led to a reduction in hepatic *Bmal1* and *Clock* expression, while in the stomach and duodenum the opposite effect was observed (Fig 3).

Sex-specific differences in gene expression patterns were also present. In some cases, for example, for *Bmal1 and hepatic Clock* under NF or SNF, female mice showed early (Day 3 post-shift) timepoint-specific differences that weakened over time, while males developed more persistent responses over time.

### Food intake during the lights-on phase downregulates hepatic gene expression

Similar to our observations with *Bmal1*, the expression of hepatic genes related to nutrient metabolism and inflammatory responses was generally lower in both male and female DF and SDF mice (Fig 4). Although timepoint-specific differences were present for multiple genes, not all showed consistent responses to the tested TRF regimes, suggesting a limited effect of food intake timing on hepatic gene regulation.

Unexpectedly, *Ppara* and *Srebf1*, the principal transcription factors regulating hepatic lipid metabolism [86] and previously observed to respond to TRF [47,87], showed only sporadic timepoint-specific changes (Fig 4, $^*$p). The most notable effect was downregulation of *Ppara* in SNF females at the postprandial timepoint, observed on multiple sacrifice days. The 4 h food intake delay also led to only minor changes, with some gene upregulation in SNF males over prolonged entrainment (Fig 4, $^\$$p).

However, not all nutrient metabolism genes were unresponsive to TRF. Genes encoding fatty acid synthase (*Fasn*), glycogen synthase (*Gys2*), and nicotinamide adenine dinucleotide (NAD$^+$) biosynthetic enzyme (*Nampt*) showed clear food-timing-induced expression changes, similar to observations before [47]. Expression of genes encoding phosphoenolpyruvate carboxykinase (*Pepck*) and fibroblast growth factor 21 (*Fgf21*) were modulated by TRF as well (Fig 4, $^*$p). While DF and SDF triggered some timepoint- and sex-specific responses for these genes, the most consistent effects in both sexes were observed for SNF. In contrast, *Fgf21* was strongly downregulated at food intake and/or postprandial timepoint under all TRF regimes, especially under longer entrainment. Given the role of *Fgf21* in nutrient and energy homeostasis and its known induction during fasting [88], this expression pattern may reflect an adaptation to the daily 16 h fast, persistently peaking during food anticipation, regardless of when feeding occurs within the light-dark cycle.

The response of inflammatory markers to food intake timing was less clear. *TNFa* showed an inconsistent upregulation at the food intake timepoint under SNF and SDF, and *Gpx1* displayed no timepoint-specific pattern.

Sex-specific differences followed a pattern similar to that already seen with clock genes. Females often showed early timepoint-specific differences that diminished or disappeared over time, while males showed delayed but more persistent gene expression changes (Fig 4., e.g., *Fasn, Pepck, Gys2, Nampt).* Male mice were also more likely to alter their gene expression in response to the feeding shift, especially after the 4-hour delay during the dark phase (SNF vs NF, $^\$$p, see Fig 4: *Ppara, Srebf1, Fasn, Cyp7a1, Mttp, Gys2*).

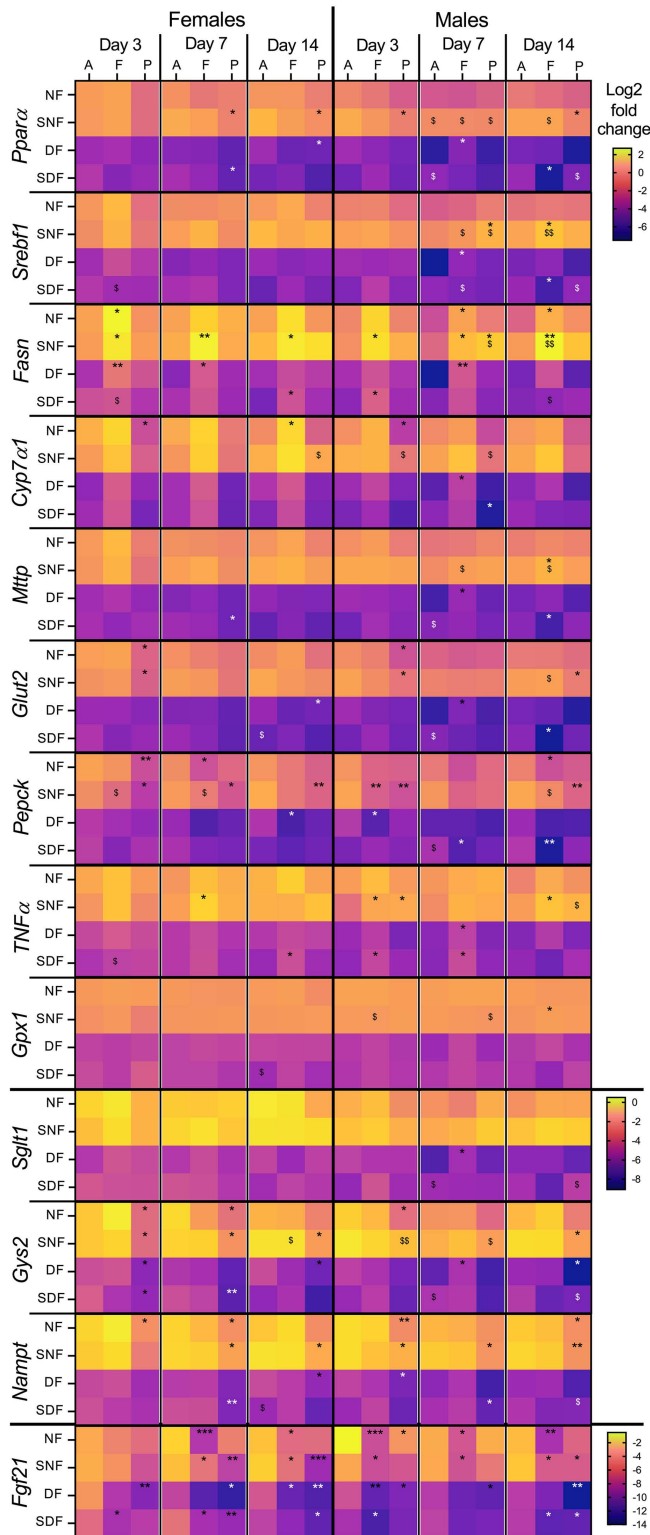

**Fig 4. Hepatic gene expression changes in male and female mice under TRF.** Sample collection timepoints shown at the top: A = food anticipation, F = food intake, P = postprandial period. Feeding conditions shown on the left: NF = night-fed (ZT12 – ZT20), SNF = shifted night-fed (ZT16 – ZT24), DF = day-fed (ZT0 – ZT8), SDF = shifted day-fed (ZT4 – ZT12). Relative gene expression values (Log2) were normalized to NF male mice on sacrifice

day 3 at the food anticipation timepoint and pooled by sex, timepoint, and sacrifice day. The mean value of each experimental group is shown as an individual tile (n = 4-5 mice). Non-parametric Kruskal-Wallis test with Dunn's multiple comparison test used to show timepoint-specific (food anticipation vs intake or vs postprandial period) differences in gene expression on each sacrifice day, *p < 0.05, **p < 0.01, ***p < 0.001. The same test was used to assess shift-specific differences (non-shifted vs shifted) at each timepoint, $p < 0.05, $$p < 0.01.

### TRF fails to elicit gene responses in the stomach

As a part of the digestive tract, the stomach is directly exposed to ingested food and has been shown to synchronize certain hormonal responses with mealtime [89]. In our study, gastric gene expression was largely unresponsive to TRF (Fig 5), similar to findings in rats [57]. The most consistent observation was an overall upregulation of gene expression (independently of the timepoints) under lights-on feeding conditions (Fig 5, DF, SDF), mirroring the pattern seen in gastric *Bmal1* (Fig 3).

Despite the absence of stable and continuous TRF-induced changes across TRF conditions, several genes exhibited similar response patterns. In DF males on Day 14, a group of genes encoding major GI hormones, their precursors or receptors (*Ghrl, Pga5, Tph1* and *Lepr*), or (anti-)inflammatory markers (*Gxp2, Il-6, TNFα*) or transcription factor *Ppara* were downregulated at the postprandial timepoint, and showed partial overlap in other timepoint-specific changes (Fig 5, *p).

Gene encoding gastrin (*Gast*), a hormone stimulating gastric acid release and gastric mucosal growth [90], showed highly variable expression patterns, with inconsistent changes across TRF regimes and sacrifice days. Although *Ppara* has been implicated in *Gast* regulation [91,92]; we observed no similarities between their gene expression responses to TRF.

An unexpected observation was the strong timepoint-specific regulation of *Pepck*, encoding a rate-limiting enzyme for gluconeogenesis. Downregulation of this gene at food intake timepoint was seen in SNF males and females, as well as in NF females and DF/SDF males after extended TRF (Fig 5). There is no evidence for gastric gluconeogenesis (it has been shown to happen in the small intestine, though [93,94]), however, *Pepck* has been proposed to play a role also in the regulation of the TCA cycle flux [95] and nutrient processing [96]. Its regulation in the stomach may therefore reflect food-driven adaptation of local metabolic functions beyond gluconeogenesis.

Food delay-induced gene expression changes were uncommon and inconsistent (Fig 5, $p). The lack of clear timepoint- or shift-specific regulation in both sexes precluded detection of any sex-specific trends.

### Duodenal genes involved in nutrient metabolism respond to food entrainment

The duodenum plays a central role in nutrient digestion and absorption; thus, it is reasonable to expect its gene expression to be responsive to feeding time and nutrient-related cues. Indeed, genes involved in glucose and amino acid uptake and/or metabolism showed clear responsiveness to TRF (Fig 6). As observed in the stomach, daytime feeding led to an overall upregulation of gene expression. Similarly, genes encoding GI hormones or their precursors (*Cck, Gip, Ccg, Tph1*) remained largely unresponsive to TRF (Figs 5,6).

Genes encoding large amino acid (*B⁰at1*), peptide (*Pept1*) and glucose (*Sglt1*) transporters, a sweet taste receptor (*Tas1r2*), an anti-inflammatory marker (*Gxp2*), a gluconeogenesis regulator (*Pepck*), and a transcription factor (*Ppara*) all showed similar timepoint-specific changes in expression (Fig 6, *p). Most of these genes were predominantly downregulated during food intake and/or postprandial timepoint under NF and SNF in both sexes. Interestingly, while daytime feeding (DF) did not always lead to timepoint-specific regulation (especially in females), the shifted DF regime (SDF) elicited similar expression patterns as NF, suggesting better alignment to food availability. It is also interesting that besides *Bmal1*, *Pepck* stood out as the only gene responding to TRF in all tested organs and both sexes. Such robust food-associated regulation may reflect a broader role of *Pepck* in nutrient metabolism, beyond its canonical gluconeogenic function.

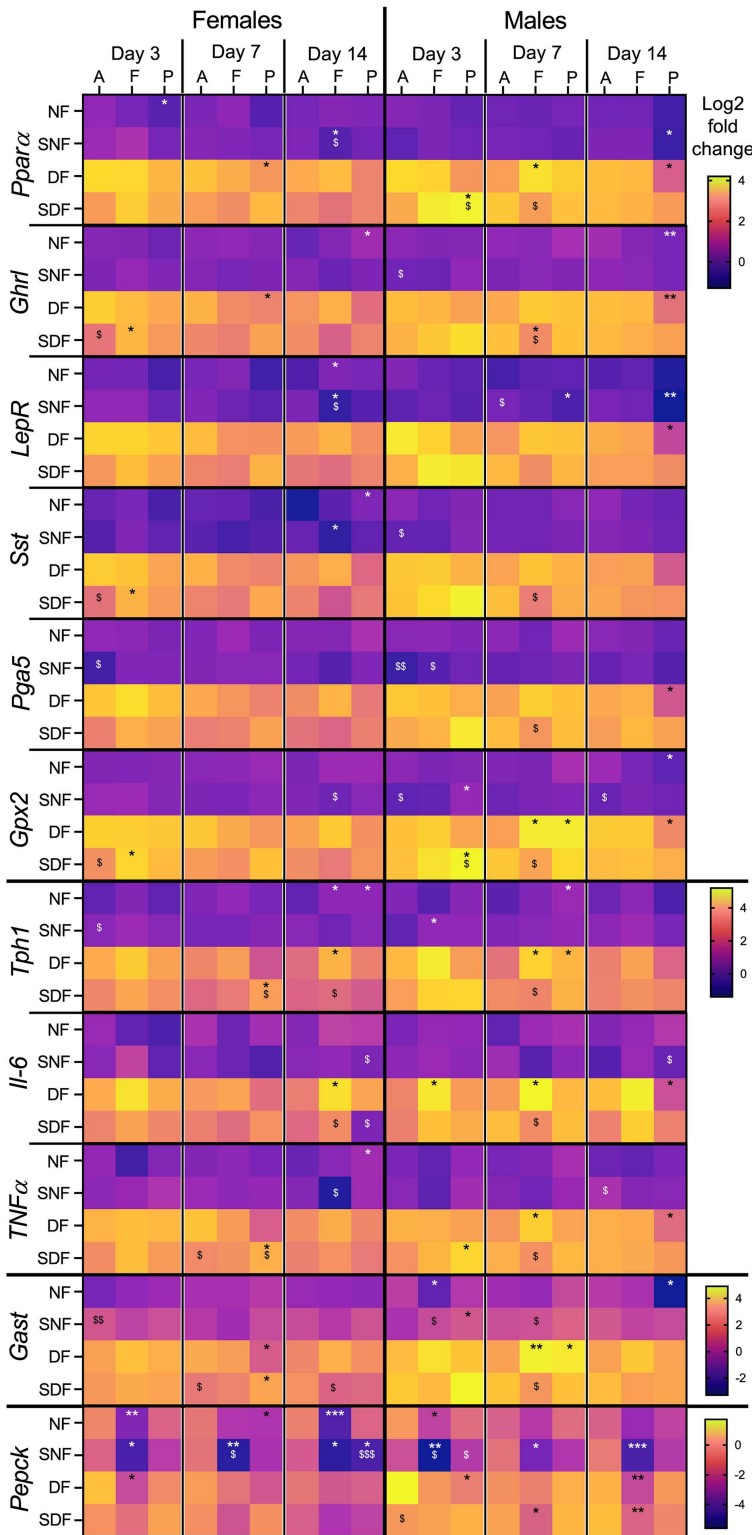

**Fig 5. Gene expression changes in the stomach of male and female mice under TRF.** Sample collection timepoints shown at the top: A = food anticipation, F = food intake, P = postprandial period. Feeding conditions shown on the left: NF = night-fed (ZT12 – ZT20), SNF = shifted night-fed (ZT16 – ZT24), DF = day-fed (ZT0 – ZT8), SDF = shifted day-fed (ZT4 – ZT12). Relative gene expression values (Log2) were normalized to NF male mice on

sacrifice day 3 at the food anticipation timepoint and pooled by sex, timepoint, and sacrifice day. The mean value of each experimental group is shown as an individual tile (n = 4-5 mice). Non-parametric Kruskal-Wallis test with Dunn's multiple comparison test used to show timepoint-specific (food antic-ipation vs intake or vs postprandial period) differences in gene expression on each sacrifice day, *p < 0.05, **p < 0.01, ***p < 0.001. The same test was used to assess shift-specific differences (non-shifted vs shifted) at each timepoint, $p < 0.05, $$p < 0.01, $$$p < 0.001.

Sex-specific changes were evident as well. Female mice often maintained or enhanced timepoint-specific gene expres-sion changes during prolonged NF, while male mice showed stronger and more uniform responses under prolonged DF and SDF. Males also appeared to be more sensitive to mealtime delay shifts, especially in the SNF group, around Day 7 post-shift (Fig 6, $p). Some early changes (e.g., anticipatory upregulation of *B⁰at1, Cck, Gip, Tph1,* and *Glut2)* were observed in SDF females, but did not persist over time.

Taken together, our findings demonstrate that food timing significantly influences gene expression in the liver, stomach and duodenum, in highly sex- and organ-specific manner. While certain genes can adapt to feeding schedules, there was little evidence of synchronized regulation across the tissues, even under prolonged TRF. The stomach remained particu-larly resistant to adapting its gene expression to TRF, despite its direct exposure to food, while duodenal responses were more robust, but still inconsistent across sexes and food intake regimes.

Generally, male mice displayed stronger and more sustained responses and were more sensitive to mealtime shifts. In contrast, females often exhibited earlier but less stable adaptation. Our study highlights the complexity of food-driven regulation in peripheral tissues and stresses the need for integrative TRF studies examining multiple organs, sexes and entrainment durations in parallel.

## Discussion

This study describes how food intake time affects weight gain, fasting glucose levels, intestinal permeability, and the expression of genes involved in the molecular clock, nutrient metabolism, and inflammatory processes in the digestive system of male and female mice. We utilized a TRF model to limit the food consumption to 8 h in a 24 h period, with the initial experimental groups starting their feeding period either at ZT12 (NF) or ZT0 (DF). After 14 days of entrainment, food intake time was delay shifted by 4 h for half of the mice (Fig 1), creating two new experimental groups: SNF (ZT16 start) and SDF (ZT4 start). NF male mice were used as a control group to evaluate the impact of shifting the feeding window and to highlight the sex-specific differences.

### TRF impact on weight gain, glucose levels, and intestinal permeability

Observed changes in physiological parameters (relative to the NF group) during the full food entrainment period have been summarized in Table 2.

Our data reveal the presence of sex-specific responses to TRF, especially regarding weight gain. When mice were fed during the lights-on phase (DF, SDF), weight gain slowed in females, but increased in males. Delay shifting the mealtime by 4 h, independently of the light-dark phase, led to a further weight gain increase in males (Fig 2A). Struggle for DF and SDF females to retain weight was also highlighted by the 16 h fasting period, leading to the lowest weights at the food anticipation timepoint (Fig 2B). Weight gain in DF and daytime-snacking male mice has been observed before [48,97,98], aligning with our observations; however, we are unaware of studies looking at weight gain in female mice under TRF. It has been observed that, under light phase feeding, the majority of physical activity still occurs during the dark phase [99–101], therefore, further measurements of locomotion and, ideally, firing rates within the central clock neurons would be needed to determine if DF and SDF females stay more active during the dark phase when no food is available to replenish the lost calories. TRF-induced sex-specific changes in food intake could also explain the observed differences, measuring individual consumption would be necessary to properly interpret the weight gain data reported here.

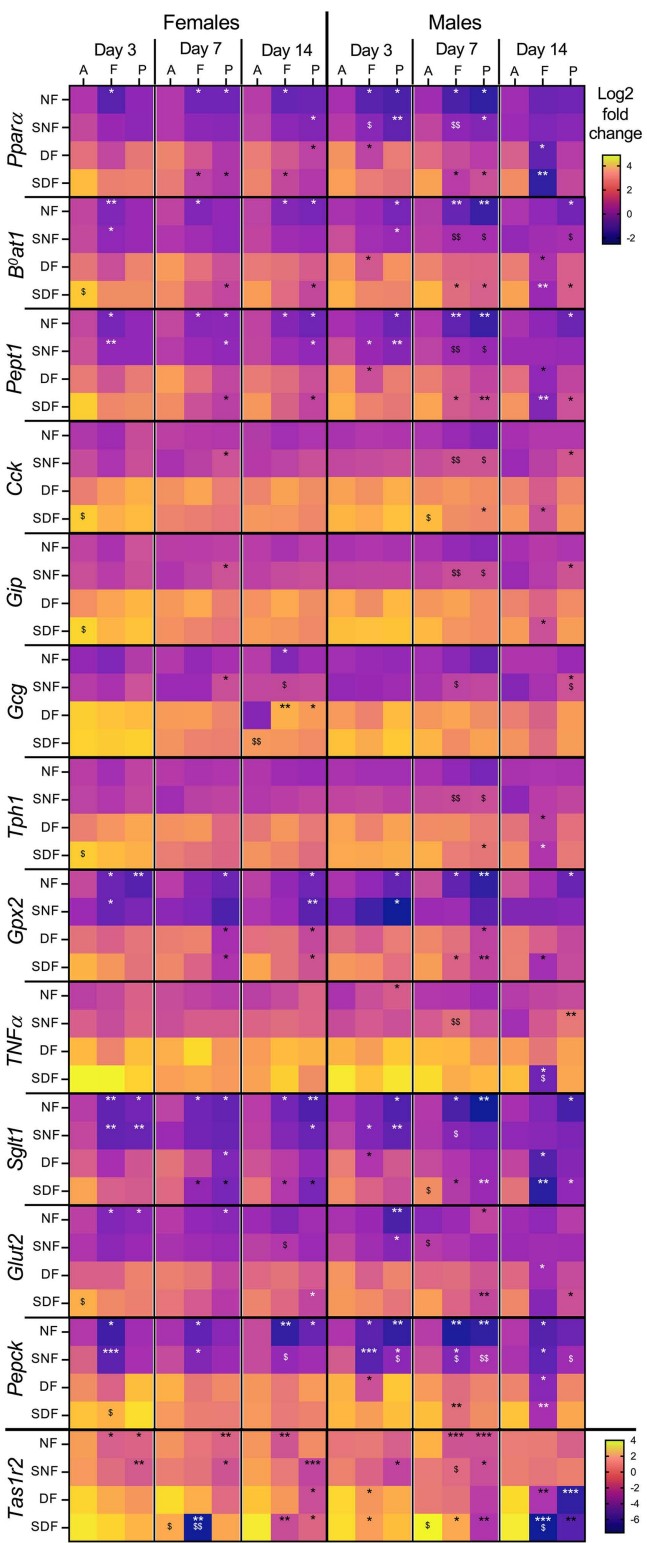

**Fig 6. Duodenal gene expression changes in male and female mice under TRF.** Sample collection timepoints shown at the top: A = food anticipation, F = food intake, P = postprandial period. Feeding conditions shown on the left: NF = night-fed (ZT12 – ZT20), SNF = shifted night-fed (ZT16 – ZT24), DF = day-fed (ZT0 – ZT8), SDF = shifted day-fed (ZT4 – ZT12). Relative gene expression values (Log2) were normalized to NF male mice on sacrifice

**Table 2. Cumulative changes in physiological parameters relative to the NF group.**

| Feeding group | Weight gain | | Fasting blood glucose | | TEER | |
|---|---|---|---|---|---|---|
| | *Females* | *Males* | *Females* | *Males* | *Females* | *Males* |
| Night-fed (NF) | highest | lowest | No change over time | | – | Increases over time |
| Shifted night-fed (SNF) | – | early↑, then - | – | early↓, then - | – | – |
| Day-fed (DF) | early↓, then - | – | – | – | – | ↓ |
| Shifted day-fed (SDF) | ↓ | ↑ | early↑, then - | – | ↓ | ↓↓ |

Fasting blood glucose levels also showed sex-specific adjustment to TRF. DF in female mice and SNF in male mice led to a significant decrease in fasting blood glucose levels, although these differences disappeared under longer entrainment (Fig 2C). This data contradicts previous observations of DF inducing higher fasting blood glucose [102], and suggests that glucose levels can normalize over long-term food entrainment. However, result interpretation would greatly benefit from measuring individual food intake and glucose levels at other timepoints.

Not much is known about TRF-driven changes in intestinal barrier function and epithelial resistance. Studies in this area often use disease models and have revealed that dark-phase TRF preserves intestinal integrity in mouse colitis model [103]. Our NF male mice also showed a significant increase in TEER over time (Fig 2D), complementing these observations. In contrast, circadian misalignment, caused by changes in the LD cycle and 'wrong-time eating', has been shown to downregulate the expression of tight junction proteins, increasing intestinal permeability [104,105] and contributing to inflammatory bowel disease (for a review, see [106]). We also observed a significant decrease in TEER under prolonged SDF of both sexes and DF of males when food- and light-driven regulation would be misaligned (Fig 2D). Further measurements of expression of tight junction proteins and their upstream effectors (such as AMPK [107]), changes in microbiota, and TEER during acute fasting-refeeding (although fasting alone seems to have no effect on ileal permeability [108]) would aid in explaining how TRF regulates intestinal barrier function.

### Hepatic, gastric, and duodenal gene expression in response to TRF

Overall, we observed three distinct gene expression responses to food intake time, summarized in Table 3. For a gene to be classified as non-responsive, it had to show only one significant timepoint- or shift-specific expression difference for the whole duration of the experiment. Interestingly, although it was possible for genes to simultaneously show both timepoint- and shift-specific changes, they seemed to occur independently from each other.

Our data (Figs 3-6) led to three major conclusions: 1) digestive organs show different rates of adaptation to TRF; 2) DF leads to an overall gene downregulation in the liver and upregulation in the stomach and duodenum when compared to NF; 3) male and female mice can exhibit opposite timepoint-specific gene expression patterns, with males more likely to also develop shift-specific responses.

Organ- and tissue-specificity in gene expression has been observed before, with surprisingly little similarity in their patterns, even if the tested genes are ubiquitously expressed and exposed to the same conditions and external cues [8,30,67,68,109–111]. This is similar to our observations, where the same gene showed timepoint-specific differences in one organ, but not in the others (e.g., *Ppara*), and genes within the same signaling pathway responded differently across

Table 3. Summary of gene expression responses to food entrainment. Data pooled from all sacrifice days and TRF regimes.

| Organ | Non-responsive genes | Timepoint-specific | | Shift-specific | |
|---|---|---|---|---|---|
| | | Females | Males | Females | Males |
| Liver | Tas1r2<br>Pept1, $B^0at1$<br>Il-17a, Il-6, Crp, Igf-1 | Clock, Pepck, Mttp (males only), Srebf1 (males only) | | | |
| | | Bmal1,<br>Glut2, Gys2,<br>Ppara, Fasn, Cyp7a1, Nampt, Fgf21,<br>TNFα | | | Bmal1,<br>Glut2, Sglt1, Gys2,<br>Ppara, Fasn, Cyp7a1,<br>Gpx1 |
| Stomach | Tas1r2, Sglt1, Glut2<br>Pept1, $B^0at1$<br>Mttp, Fasn, Cyp7a1<br>Il-17a | Bmal1, Clock, Pepck, Gast, TNFα | | | |
| | | Ppara, Tph1, Lepr, Ghrl | | Il-6, Gpx2 | |
| | | Sst | Il-6, Gpx2 | Tph1 | Ppara, Sst, Pga5, Ghrl |
| Duodenum | Cyp7a1, Mttp, Fasn<br>Sst, Ghrl<br>Il-6, Il-17a | Tas1r2, Pepck | | | |
| | | Bmal1, Sglt1, Glut2, Pept1, $B^0at1$,<br>Ppara, Gpx2 | | Gcg | |
| | | Gcg | Clock, Gip, Cck, Tph1, TNFα | Glut2 | Bmal1, Sglt1, Pept1, $B^0at1$, Ppara, Gip, Cck, Tph1, TNFα |

tissues (e.g., hepatic *Fgf21* vs. its upstream regulator [112] *Ppara*). In contrast, *Pepck* (encodes a rate-limiting enzyme in gluconeogenesis), showed a consistent timepoint-specific pattern in all organs, especially under SNF (Figs 4–6), suggesting some degree of coordination in food-driven responses. Overall, our findings indicate that TRF alone does not synchronize timepoint-specific gene expression across the digestive system, at least not under the tested conditions.

A major impact of TRF, spanning across two organs and all tested genes in both sexes, was the overall directional change in baseline gene expression: an overall upregulation in the stomach and duodenum and downregulation in the liver under DF and SDF (Figs 4–6). This observation aligns with another study [111] that observed similarities between TRF-specific up- and down-regulation patterns in murine stomach and jejunum, but not in liver. However, as more focus is put on phase shifts and rhythm dampening, such responses have been rarely described and discussed in the literature, although they have been observed (see gene expression patterns shown in [67])

Some of these effects could stem from fasting, which also alters gene expression in an organ-specific manner [113], however, the contribution of food timing *per se* remains unknown. Further studies under constant conditions, applying acute fasting and TRF, would be necessary to distinguish fasting- and TRF-driven responses.

Although sex-specificity in circadian behaviors has been described (reviewed in [114]), sex-specificity in responses to feeding time (or any other peripheral Zeitgeber, for that matter) have been understudied. Studies profiling metabolites have identified sex-specific differences under TRF and other diets, especially in the liver [67,115,116]; other work has shown sex-specific metabolic, weight, energy expenditure, and appetite regulation during fasting and refeeding [117]. Our study extends these findings, showing not only overlapping timepoint-specific responses in both sexes but also marking sex-specific differences how gene expression profiles adapt over time and in response to specific TRF regimes. Long-term TRF studies with shift reversals and repetitions could further characterize the stability and flexibility of these sex-specific adaptations and their potential health implications, especially for human populations with irregular food intake patterns (e.g., shift workers and frequent overseas or long-distance travellers).

## Limitations of the study

Lack of circadian sampling. A substantial limitation of our study is the absence of high-resolution circadian sampling, preventing an assessment and characterization of rhythmic parameters such as amplitude and phase. While this design choice limits conclusions about endogenous circadian regulation, our goal was to focus on food-driven responses under TRF, using three biologically relevant timepoints aligned to key feeding-related states (anticipation, intake, and postprandial period). A strategic alignment of these states across different TRF regimes to shared ZTs allowed us to distinguish

feeding-driven gene expression from light-driven circadian cues, even with limited sampling. While this approach does not substitute for full circadian profiling, it was successful in providing novel insights into the adaptation of peripheral gene expression to feeding schedules *in vivo*.

Impact of oestrous cycle. Estrous cycle staging was not performed in this study due to logistical constraints. However, previous studies have shown that female mice do not exhibit greater variability than males in physiological, behavioural, or molecular traits, suggesting that estrous cycle monitoring may not be necessary for detecting robust sex-specific differences in these contexts [118–120].

Daily cage transfers. A potential limitation of our study is the daily cage transfer during the TRF, which could have introduced mild handling-related stress, particularly in male mice, due to the re-establishment of social hierarchies. While elevated corticosterone levels and anxiety-like behaviours have been reported with frequent cage changes [121], efforts were made to minimize stress by using consistent "food" and "empty" cages throughout the week and by transferring portions of nesting material and enrichment objects [122]. We prioritized this approach to reduce variability in food intake measurements caused by spillage, which can be substantial in young mice [123]. Nonetheless, we acknowledge that stress responses may have influenced certain physiological outcomes and should be considered when interpreting the results.

Ambient temperature. Mice in this study were housed at standard laboratory conditions (22 ± 1 °C, 55–60% humidity). While these temperatures are common in research settings, they are below the thermoneutral zone for mice [124], which may influence metabolic homeostasis and the regulation of gene expression rhythms [125] in a sex-specific manner [126]. However, thermoneutrality has been reported to vary between the light and dark phases by approximately 4 °C [127], implying that achieving consistent thermoneutral conditions throughout the LD cycle would require dynamic temperature regulation.

Small group sizes. We used 5 mice per timepoint per sacrifice day per sex. Although the chosen sample size was small, post hoc tests (effect size/Cohen's F) for randomly chosen parameters revealed enough power to detect large effects ($q > 3.9$) and sufficient power to detect moderate effects ($3.9 > q > 2$). However, to detect small effects ($q < 2$), our chosen statistical tests could have been underpowered, suggesting that our study could have benefited from larger group sizes.

Overall, our study demonstrates the complexity of TRF responses, which are sex, organ- and gene-specific, and depend on the duration of food entrainment. It also underscores the importance of including female subjects in TRF and other peripheral entrainment studies. To further unravel the interplay between food-, fasting- and light-driven contributions to food-entrainment, future studies under constant conditions with higher sampling frequency and inclusion of ad libitum-fed controls would be necessary. These would paint a more complete picture of how food intake timing influences peripheral gene expression, even in the absence of circadian rhythmicity assessment.

## Acknowledgments

We thank Single Cell and High Throughput Genomics Platform at Monash Health Translation project, especially Dr. Sen Wang, for performing the IFC analysis. We also thank Prof. Frédéric Gachon from the Institute of Bioscience at University of Queensland for the constructive exchange regarding experimental approaches and methodology.

## Author contributions

**Conceptualization:** Lalita Oparija-Rogenmozere, Madeleine R. Di Natale, Rachel McQuade, John B. Furness.

**Data curation:** Lalita Oparija-Rogenmozere, Madeleine R. Di Natale, Billie Hunne, Ada Koo, Mitchell Ringuet, Therese E. Fazio Coles, Linda J. Fothergill, Rachel McQuade.

**Formal analysis:** Lalita Oparija-Rogenmozere, Madeleine R. Di Natale, Mitchell Ringuet, Linda J. Fothergill, Rachel McQuade, John B. Furness.

**Funding acquisition:** Lalita Oparija-Rogenmozere.

**Investigation:** Lalita Oparija-Rogenmozere, Madeleine R. Di Natale, Billie Hunne, Ada Koo, Mitchell Ringuet, Therese E. Fazio Coles, Linda J. Fothergill, Rachel McQuade.

**Methodology:** Lalita Oparija-Rogenmozere, Madeleine R. Di Natale, Billie Hunne, Ada Koo, Mitchell Ringuet, Therese E. Fazio Coles, Rachel McQuade.

**Project administration:** Lalita Oparija-Rogenmozere, Madeleine R. Di Natale, John B. Furness.

**Resources:** Rachel McQuade, John B. Furness.

**Supervision:** Lalita Oparija-Rogenmozere, Madeleine R. Di Natale, Rachel McQuade, John B. Furness.

**Validation:** Lalita Oparija-Rogenmozere, Madeleine R. Di Natale, Billie Hunne, Ada Koo, Mitchell Ringuet, Therese E. Fazio Coles, Linda J. Fothergill, Rachel McQuade, John B. Furness.

**Visualization:** Lalita Oparija-Rogenmozere, Madeleine R. Di Natale, Rachel McQuade.

**Writing – original draft:** Lalita Oparija-Rogenmozere, Madeleine R. Di Natale, Rachel McQuade.

**Writing – review & editing:** Lalita Oparija-Rogenmozere, Madeleine R. Di Natale, Billie Hunne, Ada Koo, Mitchell Ringuet, Therese E. Fazio Coles, Linda J. Fothergill, Rachel McQuade, John B. Furness.

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
