## [Decision Letter · Decision Letter 0]

17 Jul 2025

PONE-D-24-57515Time-restricted feeding leads to sex- and organ-specific responses in the murine digestive systemPLOS ONE

Dear Dr. Oparija-Rogenmozere,

Thank you for submitting your manuscript to PLOS ONE. After careful consideration, we feel that it has merit but does not fully meet PLOS ONE’s publication criteria as it currently stands. Therefore, we invite you to submit a revised version of the manuscript that addresses the points raised during the review process.

We look forward to receiving your revised manuscript.

Kind regards,

Nicholas S. Foulkes, D.Phil

Academic Editor

PLOS ONE

Journal Requirements:

2. Please note that funding information should not appear in the Acknowledgments section or other areas of your manuscript. We will only publish funding information present in the Funding Statement section of the online submission form. Please remove any funding-related text from the manuscript.

**Additional Editor Comments:**

While both reviewers accept the potential interest and importance of this study for studying the links between feeding time, clocks and metabolism as well as the impact of gender, they also raise substantial criticism concerning the rationale for and the limitations imposed by the experimental design, notably the limited number / frequency of sampling time points. Furthermore, there are many questions about missing experimental details, important controls that are lacking as well as weaknesses in the analysis and interpretation of the results.  This will require substantial reworking of the text and possibly the inclusion of additional data before this manuscript can be reconsidered for publication.

Reviewers' comments:

Reviewer's Responses to Questions

**Comments to the Author**

1. Is the manuscript technically sound, and do the data support the conclusions?

Reviewer #1: Yes

Reviewer #2: Partly

2. Has the statistical analysis been performed appropriately and rigorously? 

Reviewer #1: Yes

Reviewer #2: Yes

3. Have the authors made all data underlying the findings in their manuscript fully available?

Reviewer #1: Yes

Reviewer #2: Yes

4. Is the manuscript presented in an intelligible fashion and written in standard English?

Reviewer #1: Yes

Reviewer #2: Yes

5. Review Comments to the Author

Reviewer #1: The authors performed gene expression analysis and physiological experiments on peripheral tissues along the GI tract to explore adaptation of clock gene expression in response to feeding rhythm alterations. The findings are interesting and supported by their data.

1. The main assessment is based on gene expression profiling at three time points related to food intake. The rationale for this design should be clarified. In contrast, the main experimental paradigm for assessing food entrainment in peripheral tissues is a high resolution sampling scheme throughout the day (every 3-4 h for 1-2 days). While the findings related to organ-specific and sex-specific gene expression are interesting, these findings are not interpretable, in regard to the adaptation rates to meal time (Line 51). The authors may reframe the conclusions and explain their rationales and the difference with existing studies in the last section of Intro or in the first section of Results.

2. Reference related to food entrainment should be more balanced. It is unfair to conclude current research is fragmented (Lines 111-112). The work may benefit from including PMID:11161204, 12882317, 19940241, 22608008, 32475077, 33889826, 34550736, 35032416, 37267101, 37773240, 40088888

Reviewer #2: The manuscript entitled “Time restricted feeding leads to sex and organ specific responses in the murine digestive system” provides valuable insights into sex-specific differences in digestive and metabolic responses under time-restricted feeding (TRF). The topic is timely and relevant, particularly given increasing interest in circadian regulation of metabolism and its sex-specific aspects. The findings are potentially impactful for the field of metabolism. However, several methodological and interpretational concerns limit the strength of the conclusions and should be addressed to improve the scientific rigor of the study.

Major Points

1. Lack of Circadian Sampling

While the manuscript discusses clock gene expression and clock-regulated genes (e.g., Cck, Leptin), these assessments were conducted at a single time point. Since circadian gene expression is inherently rhythmic, multiple sampling points over 24 hours are required to determine rhythmicity, amplitude, and phase. The current approach does not support claims related to circadian regulation. A temporal sampling strategy would significantly strengthen the chronobiological component of this work.

2. Estrous Cycle Monitoring

Although the manuscript investigates sex differences, it does not address or control for the estrous cycle in female mice. Hormonal fluctuations across the estrous cycle can significantly affect metabolism, gene expression, and behavior. This is particularly relevant for parameters such as glucose tolerance. Without estrous cycle monitoring (e.g., via vaginal cytology), the interpretation of female data is limited, and variability may obscure true sex differences. If monitoring was not performed, this limitation should at least be acknowledged and discussed.

3. Cage Replacement and Stress

The Methods state that mice were group-housed and placed into new cages daily during food restriction to prevent feeding from spillage. Daily cage changes are known to induce stress, particularly in male mice, due to the re-establishment of social hierarchies. This stress response can influence metabolism and behavior and may differ by sex. The authors should consider and discuss the potential impact of this variable on the outcomes.

4. Ambient Temperature Not Reported

Ambient temperature was not reported in the manuscript. This is a critical oversight, especially in metabolic and circadian studies. Standard animal facility temperatures (21–23 °C) are below thermoneutrality for mice (~30 °C), leading to increased energy expenditure for thermoregulation. Importantly, thermoregulatory demands and strategies can differ between sexes, potentially contributing to the observed metabolic responses. The authors should report the housing temperature and discuss how it may have influenced their findings.

5. Group Housing and Feeding Behavior

How many mice were housed per cage? Group housing can affect individual feeding behavior, access to food, and timing of consumption, factors that may influence metabolic outcomes. This is especially relevant in TRF protocols. The information of diet type, which contains particular energy should be mentioned in the method section,

6. Statistical Analysis and Data Distribution

The manuscript states that parametric tests (e.g., t-tests, ANOVA) were used, but does not mention whether data were tested for normal distribution. Confirmation that assumptions for parametric testing were met is necessary to validate the statistical approach. Without this, the reliability of reported p-values may be questioned.

7. Control Group

The authors state that a control group with ad libitum feeding was included in the study; however, explicit data from this group are not presented in the figures or adequately discussed in the text. Specifically, body weight changes and gene expression profiles appear to be shown only for the food-restricted groups subjected to different time-restricted feeding (TRF) schedules. While pre-TRF baseline weights are used for normalization, dedicated control group data are notably absent. This omission limits the interpretation of the observed effects, particularly since food restriction in this study alters the sleep–wake rhythm, a factor that can independently influence both body weight and gene expression.

7. Sample Size and Replication

The sample size reported per time point was n = 5 per sex. This may be underpowered to detect subtle but biologically meaningful differences in gene expression or physiology, especially given variability in circadian and metabolic data. Additionally, details regarding biological replicates and experimental repeats are insufficient. The authors should clarify how many independent experiments were performed and discuss power considerations.

Specific Comments

• Abstract (lines 32–43):

This section lacks clarity in distinguishing findings by sex. It is unclear which effects were observed in males versus females. The authors should revise this passage to specify sex-specific outcomes more clearly.

• Glucose Tolerance Tests and Estrous Cycle:

The researchers didn't say whether they controlled for or noted the reproductive cycle phase of female subjects during the testing, which could be a methodological oversight. Since estrogen and progesterone modulate glucose metabolism, unaccounted hormonal variation could affect these outcomes. Without controlling for estrous cycle phase, sex comparisons in glucose tolerance should be interpreted with caution.

6. PLOS authors have the option to publish the peer review history of their article (what does this mean? ). If published, this will include your full peer review and any attached files.

**Do you want your identity to be public for this peer review?** For information about this choice, including consent withdrawal, please see our Privacy Policy .

Reviewer #1: No

Reviewer #2: No

---

## [Author Response · Author response to Decision Letter 1]

6 Aug 2025

All responses to reviewers have been compiled in one file and attached to this submission.

---

## [Editor Report · Decision Letter 1]

29 Aug 2025

Time-restricted feeding leads to sex- and organ-specific responses in the murine digestive system

PONE-D-24-57515R1

Dear Dr. Oparija-Rogenmozere,

We’re pleased to inform you that your manuscript has been judged scientifically suitable for publication and will be formally accepted for publication once it meets all outstanding technical requirements.

Kind regards,

Nicholas S. Foulkes, D.Phil

Academic Editor

PLOS ONE

Editor's comments:

My apologies for the delay in reaching a final decision. I totally appreciate the major constraints of this study that stemmed from the impact of the Covid restrictions and for this reason, the inclusion of additional large-scale experimental data is clearly not a realistic option. That said, I feel that the modifications to the Discussion as well as the "limitations" sections go a long way to spelling out the inherent weaknesses of this work and so now the reader can draw their own conclusions and measure the significance of the findings. In its new form, I feel the work is now suitable for publication.
---

## [Editor Report · Acceptance letter]

PONE-D-24-57515R1

PLOS ONE

Dear Dr. Oparija-Rogenmozere,

I'm pleased to inform you that your manuscript has been deemed suitable for publication in PLOS ONE. Congratulations! Your manuscript is now being handed over to our production team.

Kind regards,

on behalf of

Dr. Nicholas S. Foulkes

Academic Editor

PLOS ONE